# Time Budgets in Domesticated Male Icelandic Horses on Pasture Turnout in Winter and Spring

**DOI:** 10.3390/ani15213206

**Published:** 2025-11-04

**Authors:** Daisy E. F. Taylor, Bryony E. Lancaster, Andrea D. Ellis

**Affiliations:** 1Royal (Dick) School of Veterinary Studies, University of Edinburgh, Easter Bush Campus, Roslin, Midlothian EH25 9RG, UK; deftaylor1@gmail.com (D.E.F.T.); bryony.lancaster@ed.ac.uk (B.E.L.); 2UNEQUI Ltd., Research~Education~Innovation, Launceston, Cornwall PL15 8RT, UK

**Keywords:** time budgets, ethology, Icelandic horse, behaviour, pasture, circadian rhythm

## Abstract

**Simple Summary:**

Recording the behaviour of horses over 24 h is time-consuming and can be difficult during hours of darkness. A behavioural time budget is normally compiled by sampling the behaviour of individuals at specific intervals (e.g., every 5–10 min), but this can risk missing certain behaviours. In this study, a 24 h time budget of continuous behaviour over 3 h periods was conducted in winter and spring during fair weather to examine seasonal variation for a mixed-age herd of eight male domesticated Icelandic horses (seven geldings, 1 stallion). Individual behaviour and herd location were continuously observed during 3 h periods to cover 3 days (72 h) in both seasons. In spring, horses spent longer foraging, moving, and lying down, but spent less time standing compared to winter. There was little behavioural difference between adult and young horses. All horses lost body condition over winter and regained it over spring. The herd showed preferences for certain areas of the field for specific behaviours and during certain weather conditions. Grazing time (12–17 h) over 24 h in both seasons exceeded recommended minimum foraging times of 8–10 h for stabled horses to meet their behavioural needs, and fasting periods of the herd rarely exceeded 2 h.

**Abstract:**

There are few 24 h time budgets for horses, especially for domesticated horses kept at pasture. Most time budgets utilise short-term scan sampling, which can miss behaviours. This study aimed to assess the seasonal variation in continuous behaviour of domesticated Icelandic horses at pasture during winter and spring in fair weather. Eight Icelandic horses (11.25 ± 9.19 years; 7 geldings, 1 stallion) were observed in a 26 acre field. Herd location and individual behaviour were continuously observed during 3 h periods amounting to 3 × 24 h in winter and late spring, compiled over 43 days (~21 days per season). Seasonal variation in behaviour (ANOVA), body condition (RMANOVA), and age-group variation (independent *t*-test) were assessed, as well as associations between weather, time period, and habitat choice (chi-square). During spring, horses showed more foraging (+18%; *p* < 0.001), movement (+0.5%; *p* < 0.05), recumbency (+5.7%; *p* < 0.01) and less standing (−24.6%; *p* < 0.001) than in winter. Behavioural synchronicity occurred between adult and juvenile horses. Mean body condition reduced from 5.6 to 4.8 in the winter. Habitat preferences varied by daytime and season, and non-feeding periods lasted less than 2 h. The 24 h foraging activity (winter: 12.7 ± 0.4 h, spring: 17 ± 0.25 h) supported the current recommendation of 12 h/24 h for domesticated horses to meet ethological requirements.

## 1. Introduction

The horse is a social, non-ruminant grazing herbivore that has adapted in response to the selective pressures encountered throughout its evolution [1]. As endotherms, horses employ physiological and behavioural mechanisms to adapt their energy budget according to circannual fluctuations of their environment [2,3]. The flexibility and plasticity of equine behaviour have allowed exploitation of grazing niches in various habitats. However, the selective pressures of domestication have not removed the original behavioural phenotype. Certain management regimes may remove environmental stresses but restrict movement, foraging opportunities, and social contact, which can result in stereotypical behaviours [1,4]. These behaviours are thought to represent coping strategies for environmental deficiencies and indicate a concern for equine welfare [5].

An organism’s natural behavioural rhythm, measured by continuous time budget recording, is closely linked to their adaptation status and welfare [6]. Time budgets examine temporal sequences of behaviours and can also identify species-specific adaptive behaviours. This allows analysis of behavioural patterns, activity, and habitat use. Time budgets of groups or individuals can be influenced by endogenous factors (age, reproductive state, body condition) and exogenous factors (season, habitat, climate, herd dynamics, domestication) [7,8,9].

Most equine time budgets utilise direct visual observation with scan sampling of all or certain individuals. Intervals between samples range from 1 to 30 min in equine time budgets, but most studies use 5–10 min intervals [6,10,11,12]. However, long intervals increase the risk of missing behaviours, especially if short-lasting or low frequency. One previous study showed that accuracy was maintained when switching from 5 to 10 min intervals. However, this only applied to long-lasting behaviours, such as grazing and resting. Reducing ethograms to specific behaviours and using scan sampling reduces the effort of data collection, but may miss the adaptive significance of short-term behaviours [13], such as equine social interactions, sudden short flight, or stereotypic behaviours, which would be accounted for if continuous observation were utilised. However, visual observation also risks disturbance of natural behaviour and is not possible where visibility is limited, which likely accounts for the lack of 24 h visual time budgets throughout the literature.

Very few studies have compiled 24 h continuous visual observations across multiple seasons [14,15]. More recently, alternative methods have been used to remotely monitor behaviour, including Ethosys, which incorporates telemetery systems [16]; accelerometers, which quantify grazing, recumbency, and movement [17,18]; Equiwatch, which records chewing activity [19]; and Hoofstep, which quantifies feeding, resting, and active behaviours [20]. Small sound recorders were recently verified as a more accessible alternative to these devices to record grass intake behaviour [21]. However, while these technologies are highly accurate in behavioural quantification, not all have yet been applied to long-term time budget recording, and they are often limited to long-lasting behaviours like grazing or resting. Continuous visual observation remains the best method for recording all exhibited behaviour despite the effort required.

There is currently a gap in the literature for continuous 24 h visual time budgets of pastured domesticated horses across different seasons, especially non-breeding horses habituated to human contact and in regular training. Studies on pastured horses have currently been limited to breeding mares during summer [11], where results may be less applicable to non-breeding horses, due to the effects of reproductive state on metabolic rate and (often increased) foraging behaviour [2,22]. Time budgets of stabled horses are more available [23,24,25,26], yet most domesticated horses usually receive some kind of turnout, which mostly occurs during fair weather conditions. Therefore, this study aimed to examine the seasonal variation in 24 h time budgets of domesticated male Icelandic horses kept at pasture in winter and spring in fair weather conditions.

## 2. Materials and Methods

Ethical approval for this study was granted by the Veterinary Research Ethics Committee of the Royal (Dick) Veterinary School, University of Edinburgh (VERC ref: 108.18).

### 2.1. Study Design

This was a longitudinal comparative study, observing a small herd of domesticated Icelandic horses living outdoors in Shetland, UK, in winter and spring. Observation periods were continuous 3 h long, with a 24 h period composed of 8 × 3 h recordings collected over 2–3 days, which covered 3 × 24 h periods for winter (December–February) and 3 × 24 h periods in late spring (May–June). The primary researcher was also the sole observer, and therefore continuous 12 or 24 h recordings were deemed unreliable and unsafe. Observation periods were limited to fair weather conditions with wind speeds < 6 m/s.

### 2.2. Study Location

Data collection took place in one of the herd’s usual fields (26.26 acres), which comprises hill and pasture habitat in Mid Walls, Shetland, UK (60°14′ N and 1°38′ W). Shetland hill habitat primarily comprising bog and heathland dominated by heather (*Calluna vulgaris*) and *Potentilla erecta* interspersed with grass species, such as *Nordus stricta*. The hill habitat in the field was elevated and more exposed compared to the pasture habitat. This intergrated with other grassland species in pasture (*Holcus lanatus*, *Festuca rubra*, *Anthoxanthum odoratum*) [27,28]. Water was readily available from a running stream and multiple ditches and pools throughout the area. The herd was previously acclimatised to this area and was turned out there for at least a week before data collection commenced.

### 2.3. Animals and Management

A herd (defined by the Oxford dictionary as a company of domestic animals of one kind, kept together under the charge of one or more persons) of eight Icelandic horses kept at pasture in Shetland was observed in winter and spring. These horses were part of a mixed-age herd (*n* = 8, 11 ± 9 years) formed in August 2018 consisting of seven geldings and one stallion. The herd was kept outdoors year-round, with the exception of nights with prolonged rain and gale-force winds, when they were occasionally stabled. Horses were not trained, ridden, bred from, or provided with supplementary feedstuffs during data collection days. Normally, the adult horses (aged 4+ years, *n* = 5) were ridden 3–5 times per week, and juvenile horses (aged 1–3 years, *n* = 3) were halter trained monthly. Horses did not wear rugs.

### 2.4. Time Budgets

Time budgets were compiled through continuous visual observation of all horses for 8 × 3 h periods (06:00–09:00 h; 9:00–12:00 h; 12:00–15:00 h; 15:00–18:00 h; 18:00–21:00 h; 21:00–00:00 h; 00:00–03:00 h; 03:00–06:00 h), which were repeated 3 times each to result in 3 × 24 h recordings per season (winter and spring) compiled over 43 days of data collection. Night-vision binoculars (Nightfox-100V, Laserware Ltd., Bristol, UK) and a head torch were used for nocturnal observation. Horses could be easily distinguished and were continuously observed during 3 h periods using an ethogram (Table 1) to record behaviours lasting >5 s, with the observer talking quietly into a voice recorder by updating the point in time and notable behaviours (e.g.,“*01:05 h Horse X and Y lift their heads and stop grazing; 1:16 h X is grazing again; 01:18 h Y is walking*…”). The ethogram used was adapted from previous studies [12,14] and modified to record behaviours lasting over >5 s.

During observation, the observer maintained a distance of at least 5 m from the horses and walked away if approached. Three adult horses in the herd were wearing headcollars fitted with small sound recorders during observation periods as described by Taylor et al. [21], but this did not affect their natural behaviour. A pilot run of a 3 h observation period was carried out to assess the feasibility of nocturnal observation (18:00–21:00 h). Horses did not react negatively or try to interact with the observer.

### 2.5. Habitat Choice and Weather Conditions

During observation periods, the location of the herd was recorded by the observer marking their current position, arrival time, and habitat type (pasture, habitat, or gate area). Weather conditions of each observation period were obtained from a local forecast, and any deviations were noted at the time. For data analysis, a weather code was assigned to each period.

### 2.6. Body Condition Scoring

The body condition score (BCS) of each horse was measured using Henneke’s Index [29] on a scale of 1 (poor) to 9 (extremely fat) based on palpating and scoring six areas of the horse’s body from 1 to 9 and calculating the average before the start and at the end of data collection for winter and spring.

### 2.7. Data Analysis

Software SPSS (Version 22) and JASP (Version 0.9.2) were used for statistical analysis. Significance was set to *p* < 0.05. Datasets were tested for normality (Shapiro–Wilk). Non-parametric tests were used when appropriate. Measured variables included body condition scores, durations of visually observed behaviours, weather conditions and habitat choice of the herd during 3 h observation periods. Time budgets were compiled for each horse using the proportion of time spent on behaviours for each period. Descriptive statistics were carried out to present results using means and standard errors for all measures according to season and age group of horses (where relevant). An analysis of variance (ANOVA) assessed seasonal variation in behaviours for each time period with Tukey post hoc tests. A paired *t*-test examined seasonal variation in total (24 h) time budgets. An independent *t*-test assessed differences in time budgets between mature and juvenile horses for each time period across both seasons. A repeated-measures ANOVA (RMANOVA) with Bonferroni post hoc tests assessed variation in body condition scores (BCS) at the start and end of both seasons.

A chi-square test was used to check the association between habitat choice and weather conditions.

## 3. Results

### 3.1. General Observations

In both seasons, horses spent most of their time foraging, and some foraging behaviour was observed within every replicate of the 3 h observation periods. Coprophagy was observed in winter (classified under “other” behaviour), and horses pawed at frost/snow-covered ground to reveal grass. The stallion’s behaviour was synchronous with geldings during winter. In spring, the presence of mares in a nearby field increased the stallion’s movement compared to the other horses. The stallion’s behaviour did not seem to affect the rest of the herd. Therefore, for pairwise seasonal comparison of average behaviour times per period and per 24 h, the stallion’s data was omitted.

### 3.2. Time Budgets

#### 3.2.1. Overall Time Budgets

Data was amalgamated to show average 24 h time budgets for each horse (*n* = 7, stallion omitted), as shown in Table 2. In spring, foraging, locomotion, recumbency, and ‘other’ behaviours were significantly higher, while standing was significantly lower compared to winter.

Data of individual horses was amalgamated to show the average visual time budget (*n* = 7, stallion omitted) (Figure 1). In spring, grazing showed a crepuscular pattern (sunrise: 03:36–04:16 h, sunset: 21:50–22:36 h). Recumbency mainly occurred after midnight and decreased towards dusk. In winter, horses foraged less and rested more, especially from midnight until midday. Resting mainly occurred while standing, but horses were recumbent around early morning and after sunset (14:57–16:58 h). Foraging started to increase at dawn (07:20–09:11 h) and peaked around dusk. In both seasons, most social behaviour occurred between midnight and midday.

#### 3.2.2. Variation Between Age Groups

Preliminary time budgets between adult (winter: *n* = 5, 17 ± 7 years; spring: *n* = 4, 19 ± 2.6 years) and juvenile horses (*n* = 3, 2 ± 1 years) were similar across observation periods in both seasons. There was no significant difference in mean 24 h foraging times between age groups in both seasons. However, there were differences in behaviours between individual time periods (Figure 2).

On winter mornings from 06:00 to 09:00 h, juvenile horses spent longer engaged in social behaviours (Mann–Whitney *U*-test; *U*= 0, *p* = 0.036) and movement (independent *t*-test; *t*(6) = −2.56, *p* = 0.043). In spring from 06:00 to 09:00, young horses spent longer moving (*t*(5) = −2.89, *p* = 0.034), while adult horses foraged longer (*t*(5) = 3.21, *p* = 0.024). However, from 15:00 to 18:00 h, juvenile horses foraged longer than adults (*t*(5)= −3.06, *p* = 0.028). From 00:00 to 03:00 h in spring, juvenile horses foraged longer (*t*(5) = −6.40, *p* = 0.001), whereas adults spent more time standing (*t*(5) = 5.04, *p* = 0.004).

### 3.3. Body Condition

Body condition scores (BCS) of horses changed significantly throughout winter and spring (repeated-measures ANOVA; F = 27.05, *p* < 0.001). BCS decreased over winter by 0.7 points from moderate to moderately thin (*p* < 0.001, CI 95% [0.38, 0.93]) before increasing by 0.4 points and returning to moderate by spring (*p* < 0.017, CI 95% [−0.77, −0.03]) and increasing to scores similar to the start of winter (Figure 3).

### 3.4. Herd Location

Herd location was recorded for each observation period as shown in Figure 4.

In winter, horses congregated around the gate from 06:00 to 12:00 h before moving into pasture for afternoon grazing and subsequently hill habitat for nocturnal rest. On spring mornings, horses arrived earlier at the gate area from 00:00 to 06:00 h and also left earlier for grazing in pasture and hill habitat. Daytime grazing and rest occurred in pasture, and horses foraged out towards hill habitat in the evening, where they would rest. In spring, the stallion lingered around the southeast fence line while watching mares.

A weather code assigned to every observation period showed a significant association between habitat choice of the herd (chi-square; *X*^2^(384) = 81.9, *p* < 0.001). During light precipitation, horses gathered by the gate. In clear conditions, horses spent more time in hill habitat unless temperatures were >10 °C; then pasture was preferred. Light snow/ice cover did not affect habitat preferences.

## 4. Discussion

The aim of this study was to record time budgets of domesticated male Icelandic horses on a large-scale pasture turnout during winter and spring. Most domesticated horses are turned out on small paddocks, in small groups (often same sex), and for shorter periods of time, but there are no studies to record domesticated horses’ behaviour when turned out in a semi-feral manner [30].

Due to budgetary and safety limitations, 3 h observation periods were conducted and repeated three times, so data was collated over multiple days and relied on averages per period. However, observations were continuous to ensure increased accuracy and reduce the risk of missing short-lasting behaviours, which may occur when scan sampling [12]. Carrying out shorter but more frequent observation periods did not disturb the herd since they were habituated to human contact and tolerated being approached even when recumbent. The use of sound recorders on three horses in the herd verified visual observations of grazing as discussed in Taylor et al. [21].

This study was limited to data collection during fair to moderate weather (wind speed < 6 m/s and light rain) conditions, which often reflect conditions when domesticated horses are turned out. Due to the geographic location, we needed to avoid often recurring wind speeds of >50 mph. This meant that each season’s observation period was spread over approximately 20 days, which was not seen as a limitation, as standard deviation of results within a season was relatively low. It is possible that some observed behaviour was residual from previous environmental conditions, such as adverse weather, which were not recorded. In future studies, behaviour during adverse weather conditions could be of interest.

### 4.1. Time Budgets

#### 4.1.1. Foraging

Foraging times increased from 52.6% to 70% per 24 h in winter and spring, respectively, while resting decreased, which is in line with other studies on feral horses in the northern hemisphere [6,14,31]. Discrepancies between values are likely due to variations in climate, habitat, breeds, and methodology. Additionally, the effects of latitudinal variation between study locations and season on photoperiod influence circadian rhythms [3]. For example, spring foraging was higher than reported for Camargue (55–64% of 24 h) and Przewalski horses (54%) [14,16], but similar to Konik horses (67–75% of 24 h in early spring) [15]. Similarly, Icelandic horses in Iceland showed long grazing times in late summer, which were attributed to seasonal increases in appetite hormones and metabolic rates [2,31]. In some studies, different specifications between day and night may mask ultradian behaviour patterns since some define it as hours of daylight/darkness while others utilise specified time periods [6,14]. This is avoided with 24 h observations.

In this study, wintery conditions persisted until observations started in late spring, so environmental conditions may have reflected early spring. Over 24 h, feed intake times (12–17 h) in both seasons exceeded the minimum 8–10 h recommended to satisfy ethological requirements of stabled domesticated horses and related more to optimum times of 12 h and more [32].

Ultradian behavioural patterns of this herd in winter and spring showed rest periods occurring around mid-morning/midday, midnight, and before dawn, as also observed in Camargue, Przewalski, and Konik horses [14,15,33]. Crepuscular grazing patterns are shown in these feral populations across seasons [14,15,16] and were observed in the study herd during spring. Grazing times of the herd at dawn in winter were unexpectedly low. However, throughout the day, grazing times increased to peak at dusk.

#### 4.1.2. Resting

During winter, the herd mainly rested by standing, but exhibited more recumbency in spring, which was also observed in Camargue horses [14]. A preference for standing during winter may aid in reducing heat loss via conduction compared to when recumbent on cold ground [34]. In both seasons, grazing times decreased from midnight onwards as the herd rested until approximately 03:00 h. Resting after midnight was observed in Camargue and Konik herds [14,15] and a maximum 3 h rest period after midnight has been shown in domesticated broodmares at pasture and in stabled horses [11,25]. This has led to minimum recommendations that horses should not be left stabled without foraging opportunity for more than 4–5 h [32], with an optimum of no more than 3 h, both for behavioural welfare and to protect excessive acidification within the stomach [35].

In this study, it is possible that if the herd was resting towards the end of an observation period, it could continue into the next period and therefore not be recorded in its entirety. Only two single winter periods of 03:00–06:00 h and 06:00–09:00 h have the potential to show a fasting/resting period of >4 h. However, the herd was observed to spend time grazing during every observation period across both seasons, and the fully observed complete resting/recumbency periods rarely lasted >2 h.

A full 24 h continuous observation as employed previously by Vulink et al. [15] may have been more accurate, but the observers and data loggers measured grazing as gross feeding time, which included feeding and walking behaviours, and consecutive lengths of fasting periods were not reported. Other remote behavioural monitoring systems, such as EquiWatch [19], IceTag [17], Hoofstep [20], and other accelerometers [18] have successfully quantified grazing, resting, and movement behaviours over shorter time periods due to battery capacity.

In spring, ‘other’ behaviours significantly increased (+0.2%, *p* < 0.05). Higher frequencies of maintenance behaviours (e.g., self-grooming, mutual grooming, rolling) in spring were likely a result of shedding the winter coat, while increased eliminations likely reflected increased forage intakes. All horses eliminated indiscriminately with the exception of marking behaviour by the stallion in accordance with previous observations of free-ranging horses in large heterogeneous environments [22].

#### 4.1.3. Effect of Age

We only had three younger horses within this small group, and overall, adult and juvenile Icelandic horses in this herd were mostly synchronous in their ultradian behaviour, especially for mean grazing behaviour. One exception was during morning observation periods, where juveniles were more active while adults rested or grazed. However, juveniles seemed to compensate by grazing for longer later in the day. Synchronicity between adult and juvenile horses was also shown in a feral Przewalski herd where yearlings and 2-year-olds behaved more similarly to adults than foals [36]. In our study, playing occurred frequently within and between juveniles and adults. Typically, younger horses play, and this behaviour decreases throughout maturation, but is observed more in males [37]. The playing seen in this study was likely due to the herd comprising males of mixed ages, which may not be representative of mixed or other herd structures, and can only be seen as pilot data due to the small sample group. Domesticated horses are often kept partially stabled and partially on pasture, where the size of the pasture has been shown to affect the activity of horses [38,39]. Management in these situations often involves less stable herds and groupings compared to the herd used in this study. Future research with variations in pasture layout, larger herd composition, and stabling/turnout regimes may assess which systems meet equid phylogenetic adaptations yet remain feasible to enforce.

### 4.2. Stallion Behaviour

Domesticated stallions are rarely turned out with other horses and often live semi-isolated lives. We decided to keep the stallion in this fully formed group to observe his behaviour. The stallion showed behavioural synchronicity with the geldings during winter, but some divergence in spring. Three mares were visible from the southern area of the field during spring, and one was likely in oestrus at some point during observations. However, they did not frequently vocalise to the stallion. The stallion often separated from the herd to vocalise to the mares and pace the fence line. In both seasons, the stallion exhibited typical marking behaviour with stud piles [40]. A “snaking” head movement has been observed in feral stallions in order to maintain herd structure [4], but was not observed in this study, which was likely due to the herd consisting of geldings.

### 4.3. Body Condition

In this study, body condition scores of the herd increased from moderately thin to moderate from winter to spring, and the herd spent more time grazing and moving compared to winter. The seasonal variation in body condition scores of Icelandic horses in this study reflected similar patterns shown in feral Przewalski horses [16], and domesticated pastured Shetland ponies, fed reduced rations in winter [41]. Feral and native-type equids adapt to seasonal variations in resource availability and quality through circannual rhythms in metabolic rate and appetite hormones. Energy expenditure in winter is reduced through decreased activity, foraging, and increased circulating leptin levels, leading to horses depleting fat reserves and losing condition [2,41,42]. Icelandic horses in this study showed significantly less movement and foraging with more standing over 24 h in winter and lost body condition. As food availability and daylight increase during spring, horses become more active due to increases in metabolic rate and increased foraging time to gain body condition and make up for body weight loss during winter and also to prepare for a potential reproductive season [2]. Despite foraging less during spring, the stallion maintained a similar body condition to geldings and grazed approximately 13.8 h/24 h.

### 4.4. Habitat Choice

In spring, the herd grazed on pasture during the daytime, which may have contained grass species of higher palatability and digestibility at this time of the year. In early spring the preference may be explained by the peak in water soluble carbohydrates (WSC) in grasses typically occurring in spring and increasing towards early evening, which results in increased palatability [42]. On winter mornings, horses grazed on pasture during low temperatures on bright mornings, which may be due to freezing nights increasing the WSC content of forage [43].

Upland hill habitat was preferred for nocturnal resting for the study herd, which may allow increased vigilance, as observed in Przewalski horses [12]. This habitat was elevated by 70–90 m above sea level and contained heathers, herbs, and some rough grasses. In winter, the herd exhibited more recumbent behaviour on pasture habitat, which may have been due to this area being lower and less exposed to harsh weather conditions (50–60 m above sea level) compared to the hill habitat. However, only data from fair to moderate conditions could be gathered, and there are likely further links between behaviour, habitat choice, and weather conditions that could not be seen. The herd has been previously observed from a distance during adverse weather (heavy wind and rain) to seek sheltered areas behind stone walls or less exposed and elevated areas, and stand with their backs to the wind, which is typical of free-ranging herds [40]. It could therefore be hypothesised that grazing behaviour is suspended during adverse weather. Future research on the effect of such weather conditions on time budgets needs to confirm this. In this study, during light rain, horses congregated in the gate area but did not use stone walls for shelter. This was likely due to previous experiences of being fed there rather than waiting to go into the stable, which was not a regular occurrence. Toleration of a certain degree of adverse weather without seeking shelter may also reflect the insulative properties of their winter coats and body condition built up over the year [2,34].

## 5. Conclusions

The male domesticated Icelandic horses in a large, heterogeneous pasture used in this study exhibited seasonal variation in behaviour and physiology similar to observations of feral horses. The 24 h time budgets highlighted increases in time spent foraging (+18%, *p* < 0.001), moving (+0.5%, *p* < 0.05), and in recumbency (+5.7%, *p* < 0.05) with less standing (−24.6%, *p* < 0.001) in spring compared to winter. During winter, horses lost body condition despite maintaining moderate grazing times. In spring, increased grazing times resulted in horses regaining body condition. Grazing times in both seasons exceeded the 8–10 h recommended to meet minimum ethological requirements in domesticated horses—confirming the optimum recommended time starting at 12 h feed intake provision for domesticated horses, evenly spread over 24 h, by Harris et al. [34]. The herd grazed during every 3 h observation period. Most non-foraging periods appeared to last no more than 2 h, which is in accordance with recommendations to avoid periods of prolonged fasting of >3 h.

## Figures and Tables

**Figure 1 animals-15-03206-f001:**
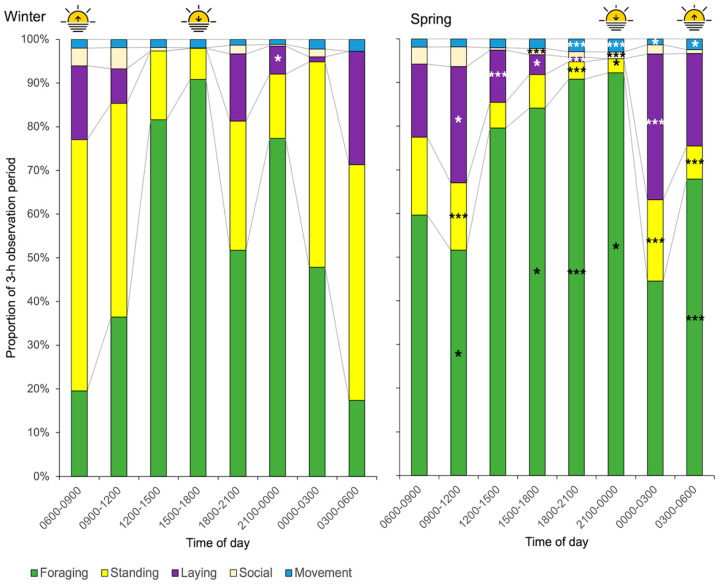
Average time budgets of seven domesticated Icelandic geldings (11.7 ± 3.7 years) at pasture compiled from 3 h observation periods over 3 × 24 h during winter and spring. Time periods that significantly differed between seasons are highlighted in spring with * *p* < 0.05, *** *p* < 0.001, ANOVA, *n* = 7 (since no lying down occurred from 21:00 to 00:00 h in spring, significance was presented in winter). Sunrise and sunset are denoted with symbols for each season.

**Figure 2 animals-15-03206-f002:**
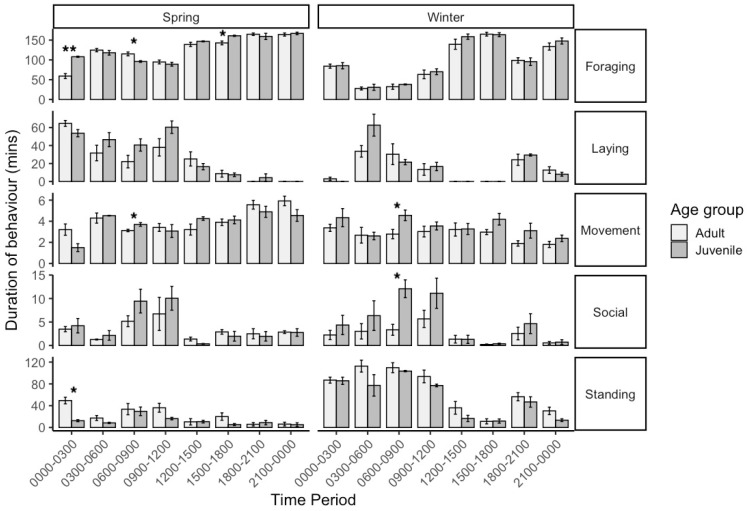
Average time budgets of adult (winter: *n* = 5, 17 ± 7 years; spring: *n* = 4, 19 ± 2.6 years) and juvenile Icelandic geldings (*n* = 3, 2 ± 1 years) at pasture compiled from 3 h observation periods over 3 × 24 h. Significant differences between age groups are denoted with * *p* < 0.05, ** *p* < 0.01 (Independent *t*-test/Mann–Whitney *U*-test).

**Figure 3 animals-15-03206-f003:**
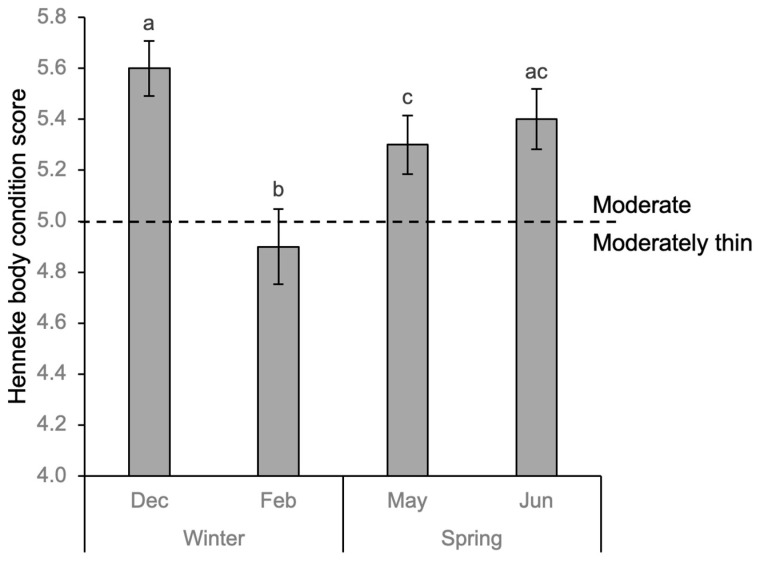
Body condition scores (BCS) on a 1–9 scale at the start and end of winter and spring for eight Icelandic horses at pasture. Vertical bars denote standard errors of scores. Different superscript letters are significantly different at *p* < 0.01 (RMANOVA).

**Figure 4 animals-15-03206-f004:**
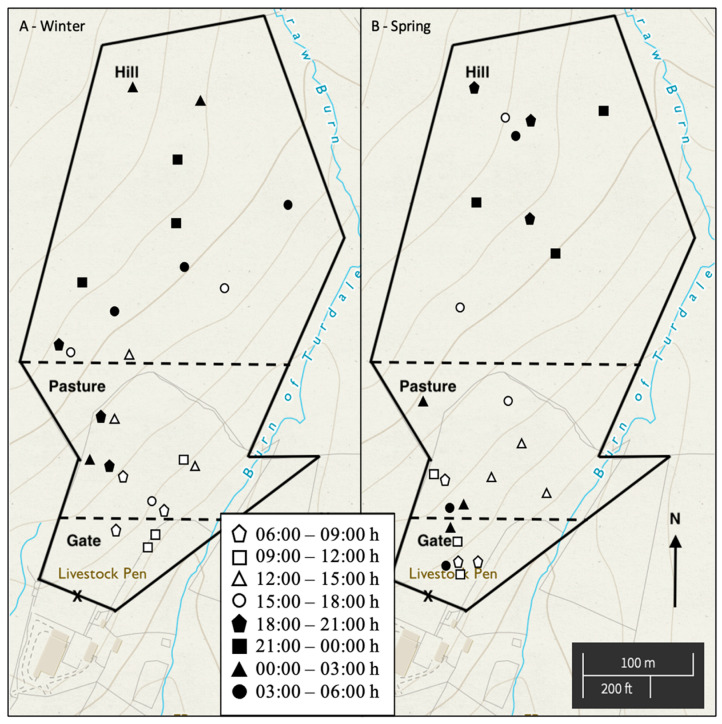
General location of a herd of domesticated Icelandic horses (*n* = 8) in (**A**) winter and (**B**) spring during 3 h observation periods. Dashed lines represent division between habitat areas (gate, pasture and hill) for data analysis. X denotes the position of the gate.

**Table 1 animals-15-03206-t001:** Ethogram utilised during 3 h continuous observation periods on eight Icelandic horses at pasture.

Behaviour	Description
Foraging	Head lowered with chewing and biting movements while standing/walking and not separated by >5 s of another behaviour.
Standing	Immobile and standing in a relaxed state with a lowered head or an alert state with head raised and pointed ears.
Lying	Sternal/lateral recumbency.
Moving	Directional movement >5 s in duration in any gait.
Drinking	Head lowered and drinking with separations <5 s.
Social	Positive/negative interactions (playing, allo-grooming and aggression, etc.).
Other	Any other behaviours (eliminations, self-grooming, and rolling).

**Table 2 animals-15-03206-t002:** Mean duration (hours) of behaviours over 24 h of seven Icelandic horse geldings at pasture in winter and spring compiled from 3 h observation periods over 3 × 24 h. (±s.e; paired *t*-test and Wilcoxon; *n* = 7, df = 6).

Behaviour	Winter	Spring	*p*-Level	Test Statistics
Foraging	12.63 ± 0.51	17.03 ± 0.25	<0.001	* t * = −9.95
Movement	0.42 ± 0.03	0.53 ± 0.02	<0.05	* t * = 2.73
Standing	8.19 ± 0.57	2.40 ± 0.42	<0.001	* t * = 31.29
Recumbency	2.20 ± 0.29	3.45 ± 0.37	<0.01	* t * = −4.07
Drinking	0.008 ± 0.003	0.002 ± 0.003	n.s.	* t * = −1.59
Social	0.50 ± 0.10	0.50 ± 0.05	n.s.	* t * = 0.24
Other	0.04 ± 0.008	0.09 ± 0.008	<0.05	* W * = 1

## Data Availability

The data presented in this manuscript is available upon request from the corresponding author.

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
