# Peer review of "Time Budgets in Domesticated Male Icelandic Horses on Pasture Turnout in Winter and Spring"

_animals, 2025, doi:10.3390/ani15213206_

Round 1
Reviewer 1 Report
Comments and Suggestions for Authors
A very interesting and thought-provoking study exploring the habits and temporal distribution of grazing behavior in a herd of domestic Icelandic horses.
Title: I suggest keeping the title focused on the preliminary nature of the data (see comments in the Materials and Methods section).
Simple Summary: Appropriate.
Abstract: I would suggest adopting a more cautious tone in the conclusions.
Introduction: Clear and comprehensive.
Materials and Methods: First of all, this is a very interesting study. Congratulations on your meticulous herd monitoring with three 24-hour recordings per season and your well-structured protocol. However, I have some doubts about this section:
Was the ethogram developed specifically for this study or was it adapted from an existing reference?
I have some concerns regarding the number and type of subjects included in the study.
On what basis was this specific herd composition chosen?
Perhaps it would have been preferable to observe a group more consistent with the typical ethological structure of the species.
The wide age range, the absence of mares, and the comparison between a single stallion and several geldings raise some concerns.
Given the presence of a herd of mares nearby, it might be interesting to analyze their behavior and possibly compare it with that of the current group.
Results:
The results are consistent with what one might expect given the composition of the herd. They do not appear to provide particularly innovative insights.
Discussion and conclusions:
I would recommend maintaining a more preliminary interpretation of the results and avoiding defining the study as significant for the breed at this stage.
Thank you anyway for the work done and for the valuable contribution in a still little-explored field.
Author Response
Thank you for all the comments which helped us to improve the paper further. See attachment for detailed replies

Reviewer 2 Report
Comments and Suggestions for Authors
The submitted manuscript titled “Seasonal Variation of Time Budgets in Domesticated Icelandic Horses at Pasture” is good work. The use of continuous observation rather than scan sampling is a major strength, allowing for a more accurate capture of nuanced behaviours that might otherwise be missed. Observing horses over 3 × 24-hour periods across seasons provides a robust dataset, especially given the 43-day span. This helps mitigate short-term variability and strengthens seasonal comparisons. The finding that foraging exceeded ethological minimums (especially in spring) is encouraging and highlights the welfare benefits of pasture-based management.
Minor suggestions:
- The study observed only eight horses, with a skewed sex ratio (7 geldings, 1 stallion). This small and unbalanced sample limits generalizability and may not reflect broader behavioural patterns across different breeds or herd structures.
- While 3 × 24-hour observations per season offer depth, spreading them over 43 days introduces variability. It's unclear how representative these snapshots are of the full seasonal behaviour.
- The study was conducted only in “fair weather,” which excludes potentially significant behavioural adaptations to adverse conditions like rain, wind, or extreme cold.
- With such a small sample, the robustness of statistical tests like ANOVA and RMANOVA may be compromised. The study doesn’t mention effect sizes or confidence intervals, which are crucial for interpreting significance.
- The classification of behaviours (e.g., “movement,” “standing”) could benefit from clearer operational definitions. Ambiguity in categorization may affect consistency and reliability of observations.
- While synchronicity between adults and juveniles is noted, the study doesn’t explore dominance hierarchies, affiliative behaviours, or conflict—key aspects of herd life that influence time budgets.
- The study mentions habitat preference variation but doesn’t detail what features of the pasture influenced these choices (e.g., shade, terrain, forage quality).
- The study reports seasonal changes in body condition but doesn’t specify the scoring method used or its reliability. Without this, it's hard to assess the validity of the claim.
- While the study shows foraging exceeded minimum ethological requirements, it doesn’t translate findings into actionable guidance for horse owners or pasture managers.
Author Response
Thank you for your feedback and for helping us to further improve the paper. Please see attachment for detailed.

Round 2
Reviewer 1 Report
Comments and Suggestions for Authors
Thank you for your response and for addressing the suggested changes.